# Polarisation Control in Arrays of Microlenses and Gratings: Performance in Visible–IR Spectral Ranges

**DOI:** 10.3390/mi14040798

**Published:** 2023-03-31

**Authors:** Haoran Mu, Daniel Smith, Tomas Katkus, Darius Gailevičius, Mangirdas Malinauskas, Yoshiaki Nishijima, Paul R. Stoddart, Dong Ruan, Meguya Ryu, Junko Morikawa, Taras Vasiliev, Valeri Lozovski, Daniel Moraru, Soon Hock Ng, Saulius Juodkazis

**Affiliations:** 1Optical Sciences Centre, Australian Research Council (ARC) Industrial Transformation Training Centre in Surface Engineering for Advanced Materials (SEAM), Swinburne University of Technology, Hawthorn, VIC 3122, Australia; 2Laser Research Center, Physics Faculty, Vilnius University, Sauletekio Ave. 10, LT-10222 Vilnius, Lithuania; 3Department of Electrical and Computer Engineering, Graduate School of Engineering, Yokohama National University, 79-5 Tokiwadai, Hodogaya-ku, Yokohama 240-8501, Japan; 4Institute of Advanced Sciences, Yokohama National University, 79-5 Tokiwadai, Hodogaya-ku, Yokohama 240-8501, Japan; 5School of Science, Computing and Engineering Technologies, Swinburne University of Technology, Hawthorn, VIC 3122, Australia; 6School of Engineering, Swinburne University of Technology, Hawthorn, VIC 3122, Australia; 7National Metrology Institute of Japan (NMIJ), National Institute of Advanced Industrial Science and Technology (AIST), Tsukuba Central 3, 1-1-1 Umezono, Tsukuba 305-8563, Japan; 8WRH Program International Research Frontiers Initiative (IRFI), Tokyo Institute of Technology, Nagatsuta-cho, Midori-ku, Yokohama 226-8503, Japan; 9CREST-JST, School of Materials and Chemical Technology, Tokyo Institute of Technology, Ookayama, Meguro-ku, Tokyo 152-8550, Japan; 10Institute of High Technologies, Taras Shevchenko National University of Kyiv, Volodymyrska Str. 60, 01602 Kyiv, Ukraine; 11Research Institute of Electronics, Shizuoka University, Johoku 3-5-1, Hamamatsu 432-8011, Japan; 12Melbourne Centre for Nanofabrication, 151 Wellington Road, Clayton, VIC 3168, Australia

**Keywords:** microlens array, laser polymerisation, graphene oxide polariser, 3D printing

## Abstract

Microlens arrays (MLAs) which are increasingly popular micro-optical elements in compact integrated optical systems were fabricated using a femtosecond direct laser write (fs-DLW) technique in the low-shrinkage SZ2080^TM^ photoresist. High-fidelity definition of 3D surfaces on IR transparent CaF_2_ substrates allowed to achieve ∼50% transmittance in the chemical fingerprinting spectral region 2–5 μm wavelengths since MLAs were only ∼10 μm high corresponding to the numerical aperture of 0.3 (the lens height is comparable with the IR wavelength). To combine diffractive and refractive capabilities in miniaturised optical setup, a graphene oxide (GO) grating acting as a linear polariser was also fabricated by fs-DLW by ablation of a 1 μm-thick GO thin film. Such an ultra-thin GO polariser can be integrated with the fabricated MLA to add dispersion control at the focal plane. Pairs of MLAs and GO polarisers were characterised throughout the visible–IR spectral window and numerical modelling was used to simulate their performance. A good match between the experimental results of MLA focusing and simulations was achieved.

## 1. Introduction

Microlens arrays (MLAs) are widely used micro-optical elements in integrated optical systems for specific applications such as laser beam homogenisation [1,2], wavefront sensors [3,4], integrated optofluidic microchips [5,6], photoelectric devices [7,8], artificial compound eyes [9,10] and 3D light field imaging [11,12]. The focal length of an MLA and image quality are dependent on the precision of its 3D surface definition, roughness and lens material. Therefore, how to precisely manufacture MLAs has always been a hot topic in academia and industry. Although a variety of fabrication strategies and processes have been proposed, such as surface-tension-effect-assisted technologies [13,14,15], ultra-high precision machining (UPM) [16,17] and photolithography [18,19], these methods are still limited by their own challenges. For example, the surface-tension-effect-assisted approach is simple and low-cost, which is suitable for mass production, but it is difficult to precisely control the surface geometry of MLAs, which is indirectly affected by temperature, pressure, wettability and processing time. The UPM technologies can produce MLAs with high-accuracy, whereas they require complex processes and are time-consuming. Lithography, as the core technology in the fabrication of integrated circuits, is a process for 2D manufacturing so there is no direct control of the required third-axis for the 3D surface morphology of the MLAs.

In this study, the femtosecond direct laser writing (fs-DLW) technique was utilised as a 3D lithography tool to precisely fabricate MLAs with the low-shrinkage SZ2080^TM^ photoresist [20]. The process of the 3D lithography via tight focusing of high-repetition femtosecond pulses is illustrated in Figure 1a. Based on such an ultra-high precision (sub-wavelength) and flexible direct polymerisation, a curvature radius of the fabricated microlens is identical to the design. Micro-optics are actively studied and routinely fabricated employing fs-laser 3D lithography and SZ2080${}^{\mathrm{TM}}$ material is a common choice for its excellent optical properties [21,22]. Polarization optics, include beam splitters, polarization routers, anti-resonant hollow-core and ring-core photonic crystal fibre waveguides, to mention a few [23].

The functionality of MLAs as refractive optical elements can be enriched when combined with diffractive elements such as a prism or grating. In this study, we used a 2D graphene oxide (GO) grating as a polariser. Laser ablation of 2–4 μm grooves with a duty cycle of ∼0.5 can define gratings as functional at comparable IR wavelengths. This further requires an assessment of GO-polarisers and MLA lens performances within the infrared (IR) spectral region.

Here, the GO grating polariser was fabricated by ablating periodic air grooves in a 1 μm GO thin film. Its polarisation performance in the IR region has been both experimentally and theoretically demonstrated with a Fourier transform infrared (FTIR) spectrometer and the finite-difference time-domain method (FDTD; Lumerical, Ansys), respectively. Additionally, a formula fitting method was adopted and mathematically demonstrated to determine the contributions resulting from absorbance and retardants to the experimentally measured IR transmittance spectra. Such an ultra-thin 2D polariser can be integrated with a fabricated MLA for polarisation and dispersion control in the IR region. They also achieved an angular tuning of dispersion in the visible range. To investigate the optical performance of the MLA and GO polariser throughout the visible–IR region, both of them were fabricated on CaF_2_ substrates rather than glass to minimise absorption.

## 2. Samples and Methods

### 2.1. Materials

A commercial hybrid organic–inorganic Zr-containing negative photopolymer SZ2080^TM^ [20] (FORTH, Heraklion, Greece) was used for photopolymerisation by direct laser writing (DLW). SZ2080^TM^ has outstanding structuring capabilities: a low shrinkage and superior mechanical robustness of the fabricated objects [24]. This photoresist consists of 20% of silica with zirconia and 80% of the polymer-forming methacryl-oxypropyl-trimethoxy-silane (MAPTMS, Polysciences Inc., Warrington, PA, USA) with methacrylic acid (MAA, Sigma-Aldrich, St. Louis, MO, United States), both having photo-polymerisable methacrylate moieties [25] with 1 wt.% Michler’s ketone (4,4${}^{\prime }$ bis(dimethyl amino) benzophenone, referred to as Bis) as the photoinitiator. The reason why Bis was chosen as the photoinitiator is its strong non-linear absorption at the used 1030 nm irradiation and high irradiation intensity. With Bis, the 3D polymerisation window is wide and this is beneficial for the efficient fabrication of large surface and volume structures (partly at the expense of reduced resolution) [26]. A detailed description of the photoresist preparation is given in ref. [25]. Samples were prepared by drop-casting on a 1 mm-thick calcium fluoride (CaF_2_ IR window 10 × 10 × 1 mm${}^{3}$ (CAFP10-10-1)) substrate with subsequent annealing at 100 ${}^{\circ }$C for 15 min in a vacuum oven without a post-exposure bake. A vacuum oven was used to promote the evaporation of the organic solvent and dehydration of the photoresist. By using this vacuum pre-bake method, volatile molecules inside the sol-gel prepolymer can be fully evaporated, thus ensuring that the prepolymer acquires a uniform hard gel form for the following laser fabrication of the 3D micro-optical structures.

The photopolymerised samples were developed for 20 min in a chemical bath of a methylisobutylketone/isopropanol (MIBK/IPA) 1:2 developer (Nippon Kayaku, Tokyo, Japan) organic solvent solution after the DLW fabrication process to remove non-polymerised material and leave the self-standing 3D structures attached to the surface of the CaF_2_ substrate. The sample was placed with an incline in the glass beaker during the development. Such a method can promote the removal of unexposed photoresist volume and lead to a cleaner micro-structure standing on the substrate. The sample was later left to dry at room temperature in ambient conditions prior to further examination.

A high-quality graphene oxide (GO) solution was synthesised by the chemical reduction of graphite via a modified Hummers method [27]. Subsequently, homogenous GO thin films with controllable thicknesses were prepared by using a vacuum filtration technique employing a polyethersulfone (PES) membrane filter with a diameter of 47 mm and a pore size of 0.1 μm (Sterlitech, Auburn, WA, USA). The thicknesses of the GO films can be controlled by monitoring the volume of the GO suspension [28]. To generate a 1-μm-thick GO thin film, 1.5 mL of a 2 mg mL${}^{-1}$ GO solution was mixed with 25∼30 mL of deionized (DI) water. Prior to filtration, the GO suspension was treated with ultra-sonication for 15 min with a Branson Sonifier to homogenise the GO solution. Then, the GO solution was poured into the filtration equipment with the PES membrane filter. Under vacuum conditions, water in the GO solution was gradually filtrated away, leaving a comparatively dried GO film on the PES substrate. Finally, the 1-μm-thick GO film was peeled off from the PES substrate and transferred onto the CaF_2_ substrate in the methanol–DI-water solution with the concentration of 80∼90%. The concentration of the organic solvent was dependent on the thickness of the GO film, and thinner films needed a more dilute solution.

### 2.2. Femtosecond Direct Laser Writing

The femtosecond (fs) laser microfabrication [26,29,30,31,32] setup, based on a Pharos (Light Conversion, Vilnius, Lithuania) laser, was integrated with the scanning Aerotech xy-stages and SCA software to control laser radiation and the scanning conditions (Workshop of Photonics, Vilnius, Lithuania). The diameter *d* and height *l* of a single photopolymerised volume pixel (voxel) was controlled by modifying the output laser power, stage-scanning speed, and objective lens focusing. In this work, an output power of the regenerative amplifier of 200 mW, a central wavelength of λ=1030 nm, a pulse duration of $t_{p}=230$ fs and a repetition rate of $f_{rep}=200$ kHz were used for both MLA and GO polariser fabrication by polymerisation and ablation, respectively.

For the MLA fabrication via direct laser polymerisation, an optical microscope objective lens (Olympus MPlanFL N 20×, Tokyo, Japan) with a numerical aperture NA=0.45 was utilised to focus the laser beam into the SZ2080^TM^ photoresist (Figure 2). In order to improve the MLA fabrication efficiency, only the outer shells of the MLA were formed with a stage-scanning speed of $v_{s}=0.1$ mm/s and pulse density of 1 × 10${}^{6}$ pulse/mm. The pulse-to-pulse separation was $v_{s}/f_{rep}=0.5$ nm. The interiors of the micro-lenses were fully polymerised using a UV light post-treatment for 5∼10 min after development. The polymerised enclosure shells with sufficient mechanical strength can avoid the leakage of unexposed regions inside the MLAs during the development processing, thus protecting the surface morphology. The outer shell was polymerised by writing several concentric rings from the bottom to top to form a plano-convex microlens based on the designed curvature radius. To ensure the shell was mechanically strong enough, the linespacing between two adjacent concentric rings was less than the lateral spot diameter of the polymerised voxel.

**Exposure conditions for 3D polymerisation**. The spot diameter of the 20×NA=0.45 objective lens was ⊘=1.22λ/NA≈2.8 μm and the designed diameter of a single microlens in the MLA was 60 μm. Therefore, 60 concentric rings with adjacent linespacing of 0.5 μm were written to form a microlens with a laser fluence of $F_{p}=0.16$ J/cm${}^{2}$/pulse or 0.7 TW/cm${}^{2}$/pulse (on the sample). The exposure dose per pulse (calculated as a linear exposure) was $D_{p}=F_{p}\times t_{p}=36.8\times 10^{-15}$ J/cm${}^{2}$. The accumulated dose over the dwell time $⊘/v_{s}$, during which the laser beam is passing across the focal diameter ⊘, was $D_{\Sigma }=D_{p}\times N_{p}\approx 206$ pJ/cm${}^{2}$, where the number of pulses $N_{p}=\left(⊘/v_{s}\right)f_{rep}=5.6\times 10^{3}$. The typical linear exposure dose of photo-lithography resists (negative and positive tone) used for a UV lamp or laser exposure is 0.1 J/cm${}^{2}$ for a high-contrast definition of patterns and a high-rate of development $R_{D}\approx 100$ nm/s [33]. Apparently, a strong non-linear contribution is required to deposit a comparable 0.1 J/cm${}^{2}$ exposure dose via the nonl-inear absorption for 3D polymerisation [34]. The first non-linear contribution, the two-photon absorption (TPA), is defined by the coefficient $\beta =\sigma^{\left(2\right)}\frac{N_{TPA}}{E_{h\nu }}$ [cm/W] with $N_{TPA}$ [cm${}^{-3}$] being the number density of TPA-absorbing molecules, $E_{h\nu }=h\nu$ [J] is the photon energy and $\sigma^{\left(2\right)}$ [cm${}^{4}$s/molecule] is the TPA cross-section measured in Goeppert–Mayer units (1 GM = $10^{-50}$ cm${}^{4}$s/molecule). The β and $\sigma^{\left(2\right)}$ are not usually provided by vendors of photo-resists/resins and are seldom measured. Apparently, β≈20 cm/TW is required for TPA polymerisation [35]. This corresponds to $\sigma^{\left(2\right)}=129$ GM considering a 0.1% molecular density of the photo-initiator in the host polymer matrix $N_{host}=\rho N_{A}/M_{host}$ [cm${}^{-3}$] for a typical resist mass density ρ=1.2 g/cm${}^{3}$ and a molar mass $M_{host}\approx 234$ g/mol [36]; $N_{A}$ is the Avogadro number. At typical $I_{p}\approx 1$ TW/cm${}^{2}$ intensities used in 3D polymerisation, βI≈20 cm${}^{-1}$, which is not a large absorption contribution via TPA. The strong absorption is defined by αd>1, where *d* is the length of absorption; e.g., for d=1μm, and $\alpha =10^{4}$ cm${}^{-1}$. Moreover, estimates of $\sigma^{\left(2\right)}\approx$ 1–10 GM are more realistic [37] considering a spectrally narrow resonant TPA band [38]. The actual contributions to the cumulative absorption coefficient $\alpha_{c}\equiv \alpha +\beta I$, along the axial depth of focus ∼2$z_{R}$, requires knowledge of the fast changing permittivity $\varepsilon ={\left(n+i\kappa \right)}^{2}$, which defines a larger absorbance due to augmented κ, which is increases via the generation of free photo-excited carriers α=4πκ/λ [22,39,40].

**Exposure conditions for GO ablation**. For the fabrication of the GO polariser, the laser beam was focused through a high-numerical-aperture (M Plan Apo HR 100×) objective lens of NA=0.9 into the d=1μm GO thin film to fabricate the grating structures with a stage scanning speed of 0.04 mm/s and a pulse density of 5 × 10${}^{6}$ pulse/mm (Figure 3). The patterned grating with a periodicity of P=4μm generated a linear GO polariser with a total area of 300 × 300 μm${}^{2}$. The focal diameter was ⊘=1.22λ/NA≈1.4μm and typical pulse energies were $E_{p}=80$ nJ. The number density of pulses per diameter was $N_{p}=\left(⊘/v_{s}\right)f_{rep}=2.8\times 10^{3}$. The ablation groove was approximately 2-μm wide and the duty cycle was 0.5 (for period P=4μm) to maximise the form birefringence of the grating/polariser pattern (Figure 3).

### 2.3. Structural and Optical Characterisation

The initial structural characterisation was made using an optical microscope (Nikon ECLIPSE LV100NPol) to confirm the survival of the MLA from the development and to observe the grating structures of the GO polariser. Additionally, the optical microscope was utilised to characterise the focal spots of the MLA in transmission mode. The 3D focal spots and longitudinal intensity distribution of the focal spots were mapped through stepping the microscope along the z−axis by 1 μm with subsequent MATLAB programming assistance.

A 3D optical profiler (Bruker ContourGT InMotion) was used to characterise the surface morphology and cross-sectional profile of the fabricated MLA and GO polariser.

Scanning electron microscopy (SEM) was used for structural characterisation of the MLA processed by laser radiation (a Raith 150TWO electron beam writer was used in the field-emission SEM mode).

Infrared (IR) transmittance spectrum of the GO polariser was measured using the microscope Fourier transform infrared (FTIR) spectrometer (Bruker V70) from 1 μm to 10 μm. An FTIR condenser was used in the microscope FTIR spectrometer to focus the broadband IR radiations on the sample in the free space. Metallic linear grid polarisers were used in a polariser–analyser setup to reveal polarisation responses of the GO polarisers.

### 2.4. Numerical Modelling

The numerical modelling of the transmittance spectra of the GO grating polariser in the IR region were simulated using the finite-difference time-domain (FDTD; Lumerical, Ansys). For the FDTD model of the GO polariser, the optical constants of the GO materials were obtained from the RefractiveIndex.INFO database [42]. The GO polariser was defined by a unit cell consisting of one GO ribbon and an air groove with a periodic boundary condition. The plane wave light source with TE or TM polarisation was placed on top of the GO polariser (3 μm away) in a normal incident direction. A transmission monitor was placed at the bottom of the GO polariser (5 μm away), perpendicular to the normal incident direction.

The light distribution in the focal region was simulated in the MATLAB program based on the Rayleigh–Sommerfeld (RS) diffraction integral [43]. Compared with Fresnel diffraction integral, the RS diffraction theory provides more accurate light diffraction predictions, because it does not assume a paraxial approximation [44]. The E-field in the focal plane $U_{2}\left(r_{2},\theta_{2}\right)$ can be calculated with the RS diffraction integral, as expressed Equation (Equation 1) [43]:(1)$$U_{2}\left(r_{2},\theta_{2},z\right)=\frac{-i}{\lambda }\;\int \int \;U_{1}^{\prime }\left(r_{1},\theta_{1}\right)\cdot \frac{e^{ikr}}{r}\cdot \cos\left(n,r\right)dr_{1}d\theta_{1},$$
where λ is the incident light wavelength, $k=\frac{2\pi }{\lambda }$ is the wave vector, *z* is the distance between the diffraction plane and the observation plane (equal to the focal length *f* for focusing), ($r_{1},\theta_{1}$) and ($r_{2},\theta_{2}$) are the polar coordinates in the diffraction plane (the plane immediately behind the microlens) and observation plane (the focal plane), respectively; $r=\sqrt{(z^{2}+{\left(x_{2}-x_{1}\right)}^{2}+{\left(y_{2}-y_{1}\right)}^{2}}=\sqrt{(z^{2}+r_{1}^{2}+r_{2}^{2}-2r_{1}r_{2}cos\left(\theta_{1}-\theta_{2}\right)}$, n denotes the unit vector normal towards the observed plane, r represents the unit vector of the r direction from point $\left(r_{1},\theta_{1}\right)$ to point $\left(r_{2},\theta_{2}\right)$, and $U_{1}^{\prime }\left(r_{1},\theta_{1}\right)$ is the E-field immediately behind the microlens. The incident wave $U_{1}\left(r_{1},\theta_{1}\right)$ is diffracted by the microlens through phase modulation, and the modified E-field by the microlens $U_{1}^{\prime }\left(r_{1},\theta_{1}\right)$ can be expressed by Equation (Equation 2) [43]: (2)$$U_{1}^{\prime }\left(r_{1},\theta_{1}\right)=U_{1}\left(r_{1},\theta_{1}\right)\cdot \begin{array}{c} \sqrt{T(r_{1},\theta_{1})} \end{array}\cdot e^{-ik\cdot \Phi (r_{1},\theta_{1})},$$
where $T(r_{1},\theta_{1})$ is the transmission distribution (which is 1 when there is no amplitude modulation), and $\Phi \left(r_{1},\theta_{1}\right)=\left[\sqrt{\begin{array}{c} r \end{array}{\left(r_{1},\theta_{1}\right)}^{2}+f^{2}}-f\right]$ is the phase modulation provided by the microlens. Finally, the light intensity on the focal plane can be calculated as the square of the E-field $I=U_{2}^{2}\left(r_{2},\theta_{2}\right)$. The lateral and axial intensity distributions were calculated for the visible spectrum; see Appendix B for the RS modelling of the axial intensity distribution at IR 1–10 μm wavelengths.

## 3. Results and Discussion

### 3.1. Structural Characterisation of the MLA

Figure 2a is the optical microscopy image of the fabricated MLA. The designed MLA consists of seven, closely-packed, microlenses to maximise its spatial filling ratio, which can reduce the light information loss resulting from the spacing among the microlenses. The designed diameter of each microlens was 60 μm with a focal length of 100 μm. According to the lensmaker’s equation with a thin lens approximation, [45], the radius of the curvature was calculated as 50.4 μm: (3)$$\frac{1}{f}=\left(n-1\right)\left[\frac{1}{R_{1}}-\frac{1}{R_{2}}\right],$$
where $n_{sz2080}\approx 1.504$, and $\frac{1}{R_{2}}\approx 0$ owing to the plano-convex design of microlenses. Based on Pythagorean theorem [46], the maximum height of the microlens was calculated as 9.9 μm.

To further characterise the dimensions and surface profile of the fabricated MLA, SEM and an optical profiler were utilised. The diameter of the fabricated MLA was measured with SEM, as shown in Figure 2b, illustrating that the diameter of the fabricated microlens matches well with the designed value of 60 μm. The slanted view SEM image also clearly shows the well-defined MLA fabricated with the 3D DLW lithography technique. The 3D topographic view (50× magnification) of the fabricated MLA is shown in Figure 2c. To accurately measure the surface profile and maximum height of the fabricated microlens with an optical profiler, a superimposed pedestal disk has also been fabricated together with a microlens, and the radius of the pedestal disk is equal to the diameter of the microlens $D_{disk}=120\;\mu$m. The microlens and pedestal disk both began at the same *Z*-position (height), thus polymerising a flat base platform with the microlens effectively protruding out from the platform. The height of the protrusions can then be measured with the optical profiler, which should ideally be the designed height, 9.9 μm $\times \;n_{SZ2080}$. Therefore, the expected height of the fabricated microlens can be achieved as illustrated in Figure 2d. The measured pedestal height is expected to be equal to half the objective depth of focus, the Rayleigh length is $z_{R}=\pi r^{2}/\lambda \approx 6\;\mu$m, where *r* is the waist (radius) at the focal point when written with the focus exactly at the resist–substrate interface. Furthermore, the surface profile of the fabricated microlens only showed a sub-wavelength surface roughness of 90 nm (min–max), smaller than the required roughness of <λ/10 for demanding optical applications at IR wavelengths. This prevents the undesired diffraction or scattering for a good optical performance. It is noteworthy that the polymerisation of MLAs had no observable polymerised modulation features due to interference caused by back-reflected light [47]. This is due to closely matched refractive indices of CaF_2_ and SZ2080^TM^.

### 3.2. Structural Characterisation of the GO Polariser

The GO grating polariser was defined by the period *P*, ribbon width *w*, and film thickness *d* (Figure 3a). Unpolarised incident light on the GO grating as TE polarisation with the electric E-field parallel to the GO grating can be coupled into a guided mode of the waveguide resulting from the guided-mode resonance [48,49,50,51], whereas its TM polarisation with the E-field perpendicular to the GO grating behaves almost identical to when there is no grating. As a result, the GO grating polariser is capable of splitting the two orthogonally oriented polarisations, coupling the TE polarisation, whilst allowing the TM polarisation to pass through. Based on the geometrical configuration of the designed GO polariser with P=4μm, w=2μm, and d=1μm, the GO grating was fabricated with the DLW technique using fs-pulses to ablate air grooves into the GO thin film. Figure 3a displays the optical microscopy image of the fabricated GO polariser; the sharp edges of the GO grating reveal good-quality fabrication with the femtosecond DLW technique. To further characterise the structures of the fabricated GO polariser, its surface morphology was measured with a 3D optical profiler (Figure 3a). The cross-sectional surface profile along the single pixel line is shown in Figure 3a. It illustrates that the periodicity, GO ribbon width, and GO film thickness of the well-defined GO grating were close to the designed values of 4 μm, 2 μm, and 1 μm, respectively. The form birefringence $\Delta n=n_{e}-n_{o}$ defined by the ordinary and extraordinary refractive indices $n_{o,e}$ of the GO grating [41] is discussed separately below and is shown for the fabricated case in Figure 3b,c.

### 3.3. MLA Focal Spot Characterisation

The focusing property of the MLA was characterised using an optical microscope in transmission mode with a white light condenser at the bottom of the optical microscope. The cross-sectional focusing intensity distribution of the MLA was captured by a 50 × NA=0.8 objective lens onto a CCD camera (Figure 4). For comparison, the corresponding theoretically simulated intensity distribution of the focal spots in the xy-plane is displayed, calculated based on the RS diffraction integral (Equation (Equation 1)) in MATLAB. The RS theory does not assume a paraxial approximation, thus providing more accurate predictions. All intensity values in Figure 4 have been normalised to their peak value. It is clear that the experimental measurements are in good agreement with the theoretical calculations. Moreover, the 3D focal spots of the MLA can be mapped by scanning the microscope with a step of 1 μm along the z−axis and stacked for 3D cross-sections using a home-made MATLAB code. Good focusing of the fabricated MLA was achieved in the lateral xy and axial yz planes of the microlens as well as the MLA. The theoretical calculation for the axial intensity distribution was also simulated with the RS diffraction integral and showed good agreement in terms of the depth of focus ∼100 μm. This demonstrates that the fabricated MLA is well-matched in terms of the design with a focal spot diameter of ∼2 μm (at an intensity maximum $1/e^{2}$-level) and a depth-of-focus (or double the Rayleigh length) of ∼13 μm (at the full width half maximum FWHM of the axial intensity) at visible wavelengths modelled using λ=633 nm. The f-number of the D=60μm lens with focal length f=100μm is $f_{\#}=f/D=1.67$, corresponding to the numerical aperture $NA=1/(2f_{\#})\approx 0.3$.

### 3.4. Form Birefringence of GO

The GO grating polariser contributes to polarisation by the transmittance *T* (and reflectance *R*) from real and imaginary parts of the refractive index (n+iκ). Specifically, absorption as well as birefringence Δn affect the optical *T* and *R* polarisation dependence.

The GO grating is a form birefringent structure with the extraordinary $n_{e}$ (along the optical axis) and ordinary $n_{o}$ refractive indices through the adjacent regions of the air grooves and GO ribbons with (sub)-wavelength widths. These widths define the volume fraction $f^{\prime }=\frac{w}{\Lambda }$ in a grating structure [52]:(4)$$n_{e}=\sqrt{\frac{n_{1}^{2}n_{2e}^{2}}{\left(1-f^{\prime }\right)n_{2e}^{2}+f^{\prime }n_{1}^{2}}};\;\;\;n_{o}=\sqrt{\left(1-f^{\prime }\right)n_{1}^{2}+f^{\prime }n_{2o}^{2}},$$
where $n_{1}$ is the refractive index of air, $n_{2e}$ and $n_{2o}$ are the extraordinary and ordinary complex refractive indices of GO, respectively, *w* is the width of one GO ribbon (2 μm), and Λ is the period of the GO grating (4 μm). A multiple location analysis of the experimentally measured optical GO constants with spectroscopic ellipsometry has been reported to precisely determine the anisotropic optical GO constants [41]. The determined extraordinary and ordinary refractive indices *n* and extinction coefficients κ of GO are plotted in Figure 3b, which were used to derive the ordinary $n_{2o}$ and extraordinary $n_{2e}$ complex refractive GO indices $\tilde{n}=n+i\kappa$. Therefore, the extraordinary $n_{e}$ and ordinary $n_{o}$ refractive indices formed by the GO grating can be calculated, then the magnitude of their difference is quantified to obtain the birefringence $\Delta n=n_{e}-n_{o}$, as shown in Figure 3c.

### 3.5. Angular Dispersion Tuning

The GO polariser also works as a diffraction grating to linearly disperse polychromatic light into its constituent wavelengths (colours), arisen from the wavefront division and interference of the incident radiation from the periodic GO grating structures. Diffraction gratings are indispensable and fundamental optical elements in applications for measuring atomic spectra in both laboratory instruments and telescopes. The MLAs are used to perform observations and imaging in the microscopic range. When these two optical elements are integrated, they enable imaging of microscopic areas while simultaneously providing spectral analysis. The dispersed light with different wavelengths can be focused at different positions by the MLA. For flexibility of the test, the GO grating and MLA were made as two separate components and assembled together for possible azimuthal rotation along the optical axis. Such an assembly provides flexibility for angular dispersion orientation (Figure 5).

The optical layout of the angular dispersion-tunable assembly is displayed in Figure 1a, where the MLA with a CaF_2_ substrate (thickness of 1 mm) was stacked on the GO grating polariser, also fabricated on a CaF_2_ substrate. The collimated polychromatic light from a white LED was dispersed by the GO grating polariser, focused by the MLA, and then captured by an optical microscope (Nikon TU Plan Fluor 50 × NA=0.80). The dispersion distribution at the focal region is shown in Figure 5, where the colours (i.e., red/green/blue) illustrate the distances between the dispersed rainbows and focal spots. Such a dispersion distribution exhibited highly centrosymmetric property. It is worth noting, that the dispersion distribution (distance of the rainbow from the focal spot) is dependent on the gap between the GO grating polariser and the MLA (here, it was 1 mm caused by the thickness of the CaF_2_ substrate). Therefore, dynamic tuning of the dispersion distribution can be achieved by increasing or decreasing the spacing between the GO grating and the MLA. Moreover, as the GO grating polariser was separated from the MLA, it was flexible to rotate the GO grating polariser, as demonstrated in Figure 5, leading to angular positioning of dispersion rainbows shown in Figure 5. Only a few angles were selected for the cases of the entire assembly (a) or the GO polariser only rotated at θ-angles. The dispersion rainbow patterns were positioned angularly and caused by the orientation of the GO polariser. The spatial separation of the red to blue colours of ∼4 μm at the focal plane was achieved with this setup. The focal spot was 2–3 times larger than the focal spot of 2 μm. There was an apparent asymmetry in the RGB colour dispersion dependent on the location of the individual lens. That asymmetry was same irrespective of the all-assembly rotation or the GO-polariser only, (a) vs. (b). This implies that the collection power of the imaging lens (of the microscope used), as well as the MLA with the pedestal, caused some colours out of the collection angle. This type of measurement could be useful for metrological characterisation of the MLA. Additionally, inspection of homogeneity in lens polymerisation could be tested if an unintended grid-like pattern is formed, as shown in Appendix A.

### 3.6. FTIR Characterisation of the MLA and GO Polariser

The MLA and GO gratings fabricated and characterised in the previous sections for the visible spectral range could be useful for the IR spectral fingerprinting region 1–10 μm, especially due to the less restrictive demand of the spatial resolution. Interestingly, the axial extent of the MLA and GO films were close to the (0.1–1)λ range for the IR domain. Here, we characterised the same optical elements for longer wavelengths (Figure 6). The transmittance of the ∼10 μm SZ2080^TM^ MLA was close to 0.5 up to 6μm and T≈0.3 for longer wavelengths up to 10 μm (Figure 6a). Polymer absorption bands from the SZ2080^TM^ resist were present; however, they would be compensated for via normalisation in spectroscopic applications. The focusing performance at IR wavelengths for the NA=0.3 microlens was calculated using Equation (Equation 1), as shown in Appendix B.

The real *n* and imaginary κ parts of the complex refractive index, $\tilde{n}=n+i\kappa$, as well as corresponding anisotropies Δn and Δκ, define the optical response of the materials through an anisotropic phase delay (retardance) resulting from birefringence and an amplitude change caused by absorbance (and its polarisation dependence linked to the material and geometrical size/shape of the sample/object). Birefringence Δn defines the retardance by $\Delta n\times \frac{d}{\lambda }$, where *d* is the sample thickness. A generic expression of the combined Malus and Beer–Lambert laws has been introduced and demonstrated to exactly fit the additive contributions of retardance and absorbance to the transmittance through a pair of aligned polariser and analyser (setup is acting in high-transmittance, opposite to the cross-polarised arrangement with no transmittance) [54]. This setup allows to determine absorption losses as well as birefringence contribution together. The principle to separate the two contributions is due to their different angular dependence.

Here, we used the fitting formula to retrieve the two contributions of retardance and absorbance to the transmittance, as expressed below:(5)$$T_{\theta }=a_{\kappa }\cos^{2}\left(\theta -b_{\kappa }\right)+o_{\kappa }+a_{n}\cos^{2}2\left(\theta -b_{n}\right)+o_{n}\equiv Abs+Ret+o_{total},$$
where $a_{\kappa }$ and $a_{n}$ are the amplitudes related to absorbance (Abs) and retardance (Ret) contributions, respectively, $b_{\kappa }$ and $b_{n}$ are the orientation dependent angles (which can be different for the two anisotropies), and $o_{\kappa }$ and $o_{n}$ are their corresponding offsets. Equation (Equation 5) indicates that the retardance caused by birefringence ∼Δn has an angular dependence twice faster on azimuthal rotation (around the optical axis) than to the absorbance (∼κ) when measured in a parallel- and crossed-polariser setup. The θ-dependence is key to separate the two contributions via the fitting method.

To investigate the absorbance and retardance contributions in the fabricated GO polariser, the transmittance spectra $T_{\theta }$ of the GO polariser was measured by the microscope FTIR spectrometer with a parallel polariser and analyser, as shown in Figure 6b. To avoid the absorption effects resulting from the substrate, both the MLA and GO polariser were fabricated on the CaF_2_ (cubic structure) substrate rather than the glass substrate, due to its IR-transparent and isotropic (no birefringence) properties. The measured IR transmittance spectra $T_{\theta }$ of CaF_2_ substrate was close to 100% (Figure 6a). During the measurement, the GO polariser was rotated by a step of $\theta =15^{\circ }$ from 0 to 180${}^{\circ }$ to investigate changes in *T* by the fit using Equation (Equation 5). On the absorption bands (at 2.94 m and 6.16 μm; the corresponding wavenumbers $\tilde{\nu }=3400$ and 1623 cm${}^{-1}$), the maximum contribution is from the absorbance *A*, while for the flat spectral range without distinct absorbance bands and at the telecom spectral window at ∼1.58μm ($\tilde{\nu }=6347$ cm${}^{-1}$), the retardance contribution caused by birefringence is recognisable. To determine the exact contributions their formulae have been fitted with a MATLAB program and the best fits are summarised in Figure 7. The markers for the bands at 2.94 and 6.16 μm show the experimental measurements of the transmittance $T_{\theta }\left(\tilde{\nu }\right)$ showing dominance of the absorption (see the best fit line) by: $T_{\theta }=0.1403\cos^{2}\left(\theta -100^{\circ }\right)+0.2476$ and $0.1904\cos^{2}\left(\theta -100^{\circ }\right)+0.5766$, respectively. The maximum of absorbance is at the orientation angle $b_{n}=100^{\circ }$.

According to Equation (Equation 5), the phase change to form birefringence (retardance) has an angular dependence twice faster than to the absorbance.Therefore, the expected retardance contribution formulae at 2.94 and 6.16 μm can also be expressed (here, the the orientation dependent angle $b_{n}=100^{\circ }$ has been assumed). The dashed-lines shown in Figure 7 illustrate that their contributions absent; they are plotted by $0.07015\cos^{2}2\left(\theta -100^{\circ }\right)+0.325$ and $0.0952\cos^{2}2\left(\theta -100^{\circ }\right)+0.67$.

In contrast, for the flat spectral range that is out of the absorption band, the retardance contribution is present. The experimentally measured *T* at 1.58 μm and the fit are plotted by $0.0471\cos^{2}\left(1.9143\theta -126.8^{\circ }\right)+0.1979$. The fitted angular varying speed was 1.9143θ, close to the expected 2θ dependence for the pure birefringent waveplate. In addition, a contribution from absorbance ($\theta -100^{\circ }$) at 1.58 μm is expected. The proportion of contribution from the retardance and absorbance could be fitted with 91.43 and 8.57%, respectively. Therefore, their amplitudes and offsets as well as the phase change of retardance $2(\theta -b_{n})$ can therefore be calculated. The contributions from retardance and absorbance have been plotted as coloured areas in Figure 7. The fits are: $0.04306\cos^{2}2\left(\theta -64.66^{\circ }\right)+0.18076$ and $0.0040365\cos^{2}\left(\theta -100^{\circ }\right)+0.017$. The sum of the two has been plotted as the final fit formula (see legend in Figure 7). Therefore, the contribution formulae of the retardance and absorbance resulting from the GO grating polariser can be fitted to the proposed method. It is worth noting that the form birefringence originates from the geometry of the grating defined by its depth and duty cycle, as discussed above. The retardance $Ret=\Delta n\times \frac{d}{\lambda }$ calculated from the form birefringence of GO is Ret≈0.11 at 2.94 μm with a thickness d=1μm.

The extinction ratio ER=10lg(TE/TM) of the GO polariser at IR wavelengths was experimentally determined (Figure 7b) and reached ∼3 at $\tilde{\nu }∼3000$ cm${}^{-1}$. This makes it sensitive to the presences of OH bands (water) in a wide range of organic and inorganic materials and composites, and also shows the hydration levels of biomaterials and food [55,56,57].In the GO family of materials, this can also show intercalated water. The degree of polarisation $DoP=\frac{T_{max}-T_{min}}{T_{max}+T_{min}}=0.27$ at the maximum extinction ratio ER at 2800 cm${}^{-1}$ (Figure 7b), at which the extinction ratio $\rho_{E}=\frac{T_{min}}{T_{max}}=0.5$ and the extinction performance of the GO grating is a linear polarizer, expressed as $\left(1/\rho_{E}:1\right)$, is 2:1.

### 3.7. FDTD Modelling of the GO Polariser in the IR Region

In terms of the phase matching condition for the guided-mode resonance [48,50], the working wavelength of the GO polariser is determined by the period *P*:(6)$$\lambda_{g}=P\cdot \left(n_{eff}-n_{air}\cdot \sin\phi \right),$$
where $\lambda_{g}$ is the guided-mode resonance wavelength, $n_{eff}$ is the effective refractive index of the guided-mode resonance depending on the duty cycle of the GO grating, $n_{air}=1$ is the refractive index of air, and ϕ is the angle of incidence; $\phi =0^{\circ }$ is the normal incidence. To investigate the performance of the GO polariser with different geometrical parameters in the IR region, the numerical modelling was simulated with the finite-difference time-domain (FDTD, Lumerical, Ansys) method with wavelengths from 3 to 10 μm. The GO polariser was modelled as one GO ribbon and an adjacent air groove with the periodic boundary condition (Figure 8).

Figure 8 displays the main results of the parameter study as the simulated IR transmittance spectra of the GO polariser with different geometrical parameters *P*, *w* and *d* under both the TE and TM incident polarisations and the difference between them. The difference indicates the polarisation efficiency of the GO polariser, which can also be considered as a filter. Figure 8 reveals that the working wavelength is dependent on the period *P* of the GO grating, which conforms to the guided-mode resonance, as expressed in Equation (Equation 6). The effect of the duty cycle of the GO grating on the performance of the polariser is shown with period *P* and GO thickness *d* at 4 and 1 μm, respectively. It is clear that the smaller the duty cycle of the GO grating (the thinner the GO ribbon), the higher the transmittance *T* modulation of the GO polariser. The influence of the GO film thickness *d* is as follows: a thicker GO film can increase the absorption of the TE polarisation as well as the TM absorption. Therefore, to select the appropriate thickness of the GO film can effectively optimise the polarisation selectivity of the GO polariser. Hence, the performance of the GO polariser is dependent on its geometrical parameters; thus, the working wavelength determined by the period of the GO grating and polarisation selectivity can be optimised by tuning the duty cycle and thickness of the grating.

Additionally, the IR transmittance spectrum of the SZ2080^TM^ MLA on a 1 mm thick CaF_2_ substrate was measured. The polymerised SZ2080^TM^ material has a relatively high transmittance from 3.5 to 5.5 μm. Therefore, the proposed GO polariser with flexible working wavelengths can be integrated with the fabricated MLA to achieve integrated polarisation control, also serving as a selective filter working together with other plain TM polariser.

## 4. Conclusions and Outlook

The fs-DLW technique was utilised as a 3D lithography tool to precisely fabricate an MLA in the low-shrinkage SZ2080^TM^ photoresist. The surface morphology of the fabricated microlenses mirrored the design with high-fidelity at high-irradiance, ∼0.7 TW/cm${}^{2}$/pulse, writing conditions. Compared with lengthy raster-scanning of the entire volume of the MLA, only the shell of the MLA was formed by the fs-DLW, whereas the inside volume was fully polymerised by UV exposure post-fabrication. This sequence significantly sped up fabrication of the MLA with prospectives for large-scale production. The experimentally measured intensity distribution of the focal spots of the MLA for visible wavelengths showed good agreement with the theoretical calculation based on the Rayleigh–Sommerfeld (RS) diffraction integral.

Additionally, a GO grating polariser was fabricated with the fs-DLW technique in a 1-μm-thick GO thin film, providing linear polarisation in the infrared (IR) region. A formula-fitting method was mathematically demonstrated to determine the contributions resulting from absorbance and retardance (caused by birefringence) in the experimentally measured IR transmittance spectrum. The polarisation performance of the GO polariser is dependent on its geometrical parameters including the period, duty cycle and thickness. The GO polariser can be integrated with the fabricated MLA for polarisation control in the IR region (working wavelength of the GO polariser), and the integrated components demonstrated a dynamic tuning of the dispersion in the visible range. Future studies are required with GO gratings to harness the strong anisotropy of IR absorption along the edges of the reduced GO where absorption can increase by up to two orders of magnitude due to the coupling between IR vibrations and free charges [53]. The high refractive index of GO at IR wavelengths could be used for sensors with reflection geometries based on surface wave generation [58] as well as on ponderomotive interactions between GO and nanoparticles [59].

## Figures and Tables

**Figure 1 micromachines-14-00798-f001:**
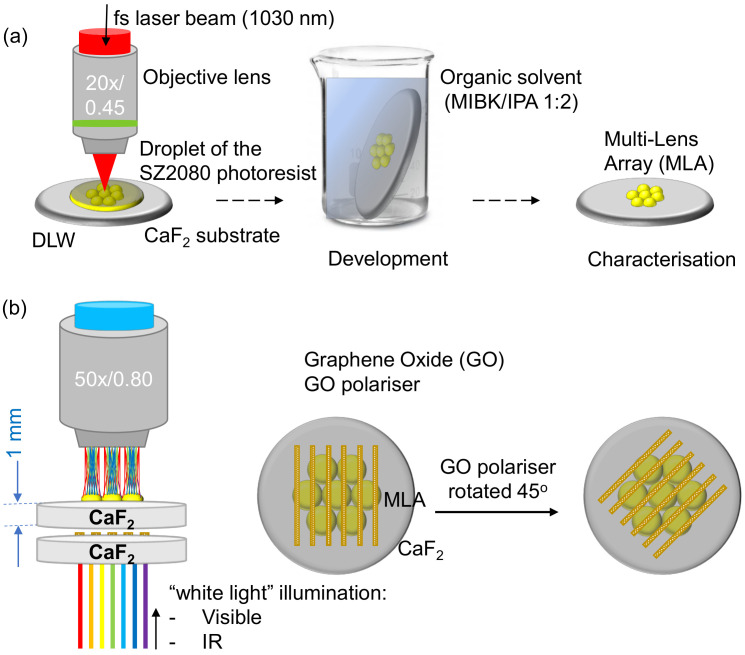
(**a**) Manufacturing steps in 3D DLW lithography. Pinpoint (maskless and selective) photopolymerisation by a tightly focused femtosecond pulsed laser beam. fs-DLW was carried out from the resist side due to the thick 1 mm CaF_2_ substrate. Removal of unexposed volume by development in organic solvent (placing the sample with an angle to promote a cleaner development). Revelation of the fabricated MLA. (**b**) Optical characterisation of the micro-optical elements (lenses/arrays and GO polarisers) in the visible and IR spectral ranges; optical elements were made on a IR-transparent CaF_2_ substrate.

**Figure 2 micromachines-14-00798-f002:**
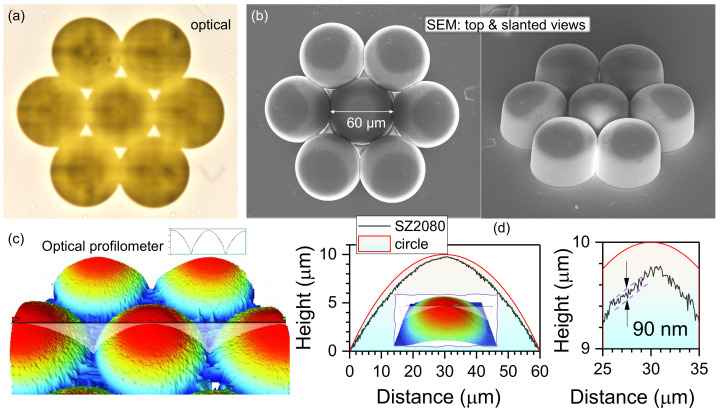
(**a**) Optical microscopy image (transmission) of the fabricated MLA. Polymerisation of the outer shell was carried out using concentric fs-DLW from larger-to-smaller diameters. (**b**) Top and 45${}^{\circ }$ tilted view of SEM images of the fabricated MLA; the diameter of the microlens is 60±0.3μm. (**c**) 3D topographic view (50×) of the fabricated MLA. (**d**) Height profile and corresponding circle curve; inset: 3D topographic view (115×). The entire height of the structure is comprised of 12μm pedestal and 10μm lens. The min–max roughness near the centre of the lens was 90±10 nm which was <λ/10 for IR wavelengths.

**Figure 3 micromachines-14-00798-f003:**
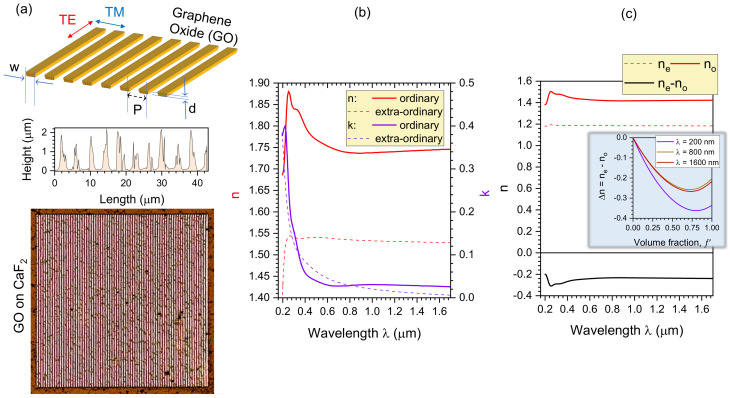
(**a**) Schematics of the GO polariser. Optical microscopy image (transmission mode) of the fabricated GO polariser 300×300μm${}^{2}$ (over the footprint of MLA) and an optical profile trace (single pixel) across the grating. (**b**,**c**) Birefringence modelling of the GO grating; see text for discussion. (**b**) The ordinary and extra-ordinary optical constants of a thick drop-cast GO layer determined by a multiple-location analysis of spatially resolved data obtained by spectroscopic ellipsometry [41]. (**c**) The ordinary and extra-ordinary complex refractive indices, as well as the birefringence $\Delta n=n_{e}-n_{o}$ of the GO grating structure, calculated by substituting the ordinary, extra-ordinary and complex refractive indices $\tilde{n}=n+i\kappa$ of GO layer from (**b**) into the standard form-birefringence formulae given by Equation (Equation 4). The inset shows Δn vs. $f^{\prime }$ dependence at different wavelengths λ; the volume fraction $f^{\prime }=1$ corresponds to a homogeneous GO film.

**Figure 4 micromachines-14-00798-f004:**
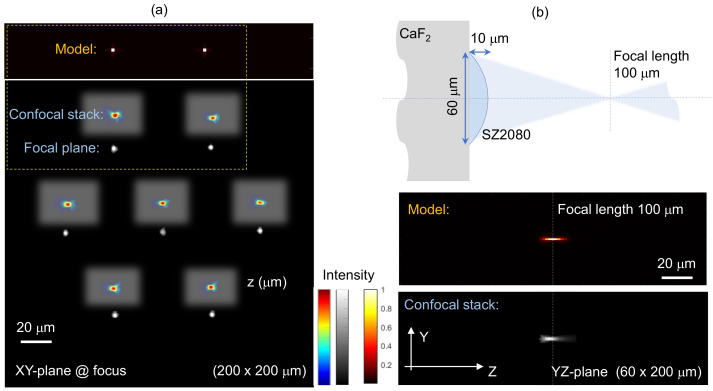
(**a**) Experimental measurements and simulations based on the model given by Equation (Equation 1) of the focal (xy-plane) and axial (yz-plane) cross-sections of the focal region intensity distribution for the MLA at a 633 nm wavelength. The cropped insets show confocally stacked intensity (normalised) at the foci. (**b**) Sketch of the lens (to scale) and the axial intensity distribution. The 3D intensity was confocally mapped through stepping microscope imaging along the *z*-axis by 1 μm and the MATLAB programming code was used to stack them for cross-sectional views. The colour bars represent the normalised intensity.

**Figure 5 micromachines-14-00798-f005:**
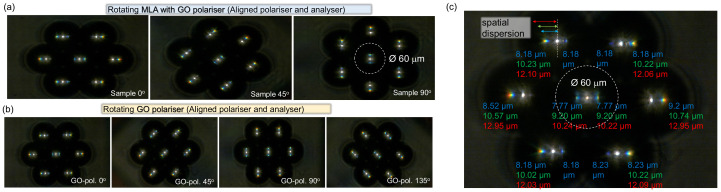
Optical microscopy imaging in the visible spectral range under white light condenser illumination. (**a**) The MLA and GO-polariser are fixed together and azimuthally rotated by the θ-angle with respect to the aligned polariser–analyser (high-transmittance mode). (**b**) The MLA is fixed and the GO-polariser is θ-rotated around the optical axis and aligned with the polariser–analyser. (**c**) RGB colour dispersion at the focal region measured by the optical microscope; the colour indicates the distances between the dispersed RGB colours and the focal spot centres. Substrates for the MLA and GO-polariser were 1-mm-thick CaF_2_. The NA of the imaging lens (Nikon Microscope TU Plan Fluor, 50×) was 0.80.

**Figure 6 micromachines-14-00798-f006:**
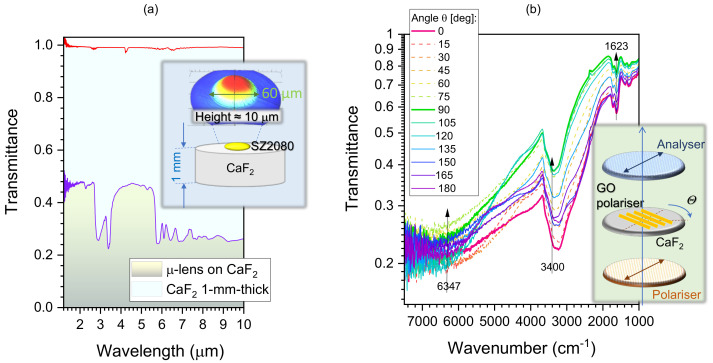
(**a**) IR transmittance spectra of the fabricated GO polariser measured by a microscope FTIR spectrometer with a parallel polariser–analyser. (**b**) Transmittance and the fitted formulae of the contributions from absorbance and the retardants at 1.58, 2.94, and 6.16 μm, as marked in the transmittance spectra. Note that *T* is plotted on a logarithmic scale to better reveal any small changes. At 3530 and 1080 cm${}^{-1}$, C-OH vibrations are present [53].

**Figure 7 micromachines-14-00798-f007:**
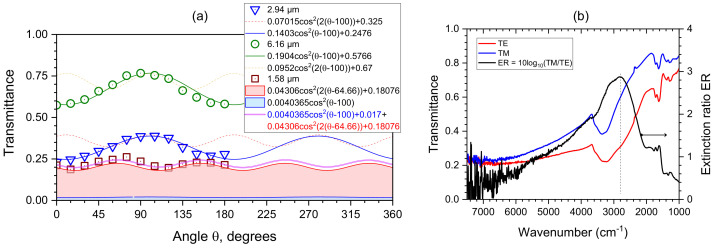
(**a**) IR transmittance spectra of the fabricated GO polariser measured by the microscope FTIR spectrometer with a parallel polariser and analyser. Transmittance was fitted considering contributions from absorbance and retardance at the 1.58, 2.94, and 6.16 μm bands as marked in the transmittance spectra (Figure 6b). The best fit formulae are shown as retrieved from the MATLAB best fit routine without rounding. See text for details. (**b**) The experimentally determined spectrum of the extinction ratio ER of the GO polariser with period P=4μm, thickness d=1μm and ribbon width w=2μm (see Figure 8 for a family of the parameter study).

**Figure 8 micromachines-14-00798-f008:**
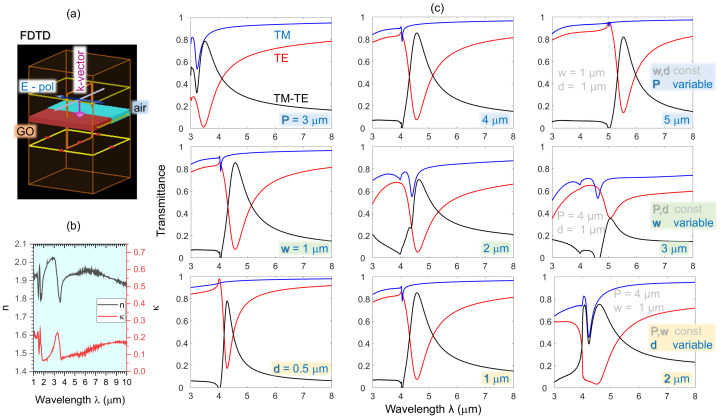
Numerical parameters P,w, and *d* for the study of the GO grating properties by FDTD. (**a**) FDTD model. (**b**) Index (n,κ) of GO used in the modelling (Lumerical database). (**c**) IR transmittance spectra from 3 to 8 μm of the GO gratings with different parameters: (**top-row**) width w=1μm and film thickness d=1μm, while the grating period *P* changes from 3, 4, and 5 μm, respectively; (**middle-row**) P=4μm, and d=1μm, while *w* is 1, 2, and 3 μm, respectively; (**bottom-row**) P=4μm and w=1μm, while *d* is 0.5, 1, and 2 μm, respectively.

## Data Availability

Data are available upon reasonable request from the corresponding author.

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
