# Peer review of "Polarisation Control in Arrays of Microlenses and Gratings: Performance in Visible–IR Spectral Ranges"

_micromachines, 2023, doi:10.3390/mi14040798_

Round 1

Reviewer 1 Report

The manuscript shows research regarding fabrication of Microlens array (MLA) by fs Direct Laser Writing (DLW) in low-shrinkage photoresist. The functionality of MLA was enhanced by integrating a grating element which was also manufactured by fs DLW in 2D graphene-oxide (GO) layer and served as polariser. The performance of MLA and GO polarizer are numerically modelled and the manufactured samples were tested for their structural and optical characterization by several techniques (optical microscopy, SEM, optical profilometer, FTIR). They were tested both in visible region and IR region up to 10 microns. The predictions from modelling are in agreement with the measured characteristics.

The main contributions are usage of enclosure shell made by DLW to enhance the speed of production (the inner parts of MLA were polymerized by post UV light exposure) and employment of the MLA in IR.

The numerical procedures and experiments are clearly presented with enough details to facilitate a possible reproduction. Conclusions follow quite naturally from the experiments' results and manuscript is easy to follow. I recommend the paper for publication without changes.

Some minor errors:

- check the formula for Phi at line 174

- Figure 4, color bar is given only for intensity in the yz plane, but not for the intensity in xy.

Author Response

Thank you for the positive evaluation of our mauscript.

Questions: Some minor errors:

  • check the formula for Phi at line 174

Answer: Thank you. Corrected.

  • Figure 4, color bar is given only for intensity in the yz plane, but not for the intensity in xy.

Answer: Thank you. Scale bar is added. All scale bars are normalised. 

Reviewer 2 Report

In this paper, a combination of fs-DLW and post-fabrication UV exposure technology is used to successfully prepare microlens arrays (MLAs) with SZ2080TM photoresist that meet the design size. The SEM image confirmed that the size of the prepared MLA was consistent with the design. The experimental measurements by using the optical microscope in transmission mode with a white light present a good agreement with the theoretical calculations. A GO grating polarizer for infrared polarization control was prepared by femtosecond laser direct writing technique on a 1μm thick GO film. The polarization performance of a polarizer depends on its geometric parameters, including period, duty cycle and thickness. The article has a certain degree of innovation, however, there are some points that I think the authors should address prior to publication, as detailed below.

1.          The introduction should fully cover the relevant research of the work done, including microlens fabrication and polarizer performance, especially in recent years.

2.          The title mentioned “Integrated Polarisation Control in Microlens Arrays”. The schematic diagram on the left of Figure 1b shows the stacking of MLAs and polarizer. Integrated devices are not found elsewhere in the paper, so more data is needed to support its representation.

3.          Data related to GO film thickness, roughness and grating detail morphology should be added in Figure 3. The white line mentioned in the paper is not drawn in Figure 3.

4.          The paper mentioned “the surface profile of the fabricated microlens showed only subwavelength surface roughness of 90 nm (min-max), which is smaller than the required roughness of < λ/10 for demanding optical applications at the IR wavelengths”. How does the roughness of the microlens affect the visible wavelengths? In reference 15 and 16, the roughness of microlens made by them is 6.98 ± 0.78nm and ~5nm respectively, the main advantages of this article?

5.          There is a big difference between the experimental data in Figure 7b and the theoretical data in Figure 8b. What causes this?

6.          In addition, this article has many formatting issues: for example, the abbreviation in line 18 appears for the first time without a full name, the formula for the number of pulses on line 114 is incorrectly written, Percentages and decimals on line 308 are not in uniform format, the size and font of the figures are not uniform, and so on. The author should seriously modify the whole manuscript.

Author Response

Answers to critical remarks:

1. The introduction should fully cover the relevant research of the work done, including microlens fabrication and polarizer performance, especially in recent years.

Answer 1.  Thank you. it is valid point. we have added recent developments. 

2. The title mentioned “Integrated Polarisation Control in Microlens Arrays”. The schematic diagram on the left of Figure 1b shows the stacking of MLAs and polarizer. Integrated devices are not found elsewhere in the paper, so more data is needed to support its representation.

Answer 2. Title is changed since integrated optical element might be considered as a single unit. What we showed is indeed performance of integrated optical element, however, the integration was not the main aim. 

3. Data related to GO film thickness, roughness and grating detail morphology should be added in Figure 3. The white line mentioned in the paper is not drawn in Figure 3.

Answer 3. Figure 3a middle panel shows the thickness, period and roughness. Text was corrected. 

4. The paper mentioned “the surface profile of the fabricated microlens showed only subwavelength surface roughness of 90 nm (min-max), which is smaller than the required roughness of < λ/10 for demanding optical applications at the IR wavelengths”. How does the roughness of the microlens affect the visible wavelengths? In reference 15 and 16, the roughness of microlens made by them is 6.98 ± 0.78nm and ~5nm respectively, the main advantages of this article?

Answer 4. Good point. since we aimed IR applications, we were able to use less tight focusing during laser polymerisation. This was partially responsible for larger surface roughness. We were not aiming at the sub-10 nm resolution which indeed is required for demanding applications at visible spectral range.  We show all details at our used polymerisation conditions. 

5. There is a big difference between the experimental data in Figure 7b and the theoretical data in Figure 8b. What causes this?

Answer 5. GO properties were taken from the a database for modelling by Lumerical, while in experiments real GO film was used and differences are expected,  FDTD has ideal structure without surface roughness. Also, in FDTD, the coherence caused effects are important since the phase of light is defined by position of the light source. In experiments, incoherent, phase randomised source was used. Qualitative match between experiment and theory is achieved and silulations explored wide parameter space. 

6. In addition, this article has many formatting issues: for example, the abbreviation in line 18 appears for the first time without a full name, the formula for the number of pulses on line 114 is incorrectly written, Percentages and decimals on line 308 are not in uniform format, the size and font of the figures are not uniform, and so on. The author should seriously modify the whole manuscript.

Answer 6. Thank you. revised and corrected. 

Reviewer 3 Report

The authors present a very thorough study. Well written, with clear figures and ample experimental parameters.

The experimental design is critically motivated and executed, and finally the conclusion is supported by the presented data, both experimental and supplemented with simulation (FDTD). The authors fs-DLW technique seems well suited for the fabrication of MLs and polarizers operating in the MWIR.

I'm happy to recommend publication of the manuscript as is!

Author Response

thank you for positive evaluation